# Robust Ranking Explanations

**Chao Chen** [1]   **Chenghua Guo** [2]   **Guixiang Ma** [3]   **Ming Zeng** [4]   **Xi Zhang** [2]   **Sihong Xie** [1]

## Abstract

Robust explanations of machine learning models are critical to establish human trust in the models. Due to limited cognition capability, most humans can only interpret the top few salient features. It is critical to make top salient features robust to adversarial attacks, especially those against the more vulnerable gradient-based explanations. Existing defense measures robustness using $\ell_p$-norms, which have weaker protection power. We define explanation thickness for measuring salient features ranking stability, and derive tractable surrogate bounds of the thickness to design the *R2ET* algorithm to efficiently maximize the thickness and anchor top salient features. Theoretically, we prove a connection between R2ET and adversarial training. Experiments with a wide spectrum of network architectures and data modalities, including brain networks, demonstrate that R2ET attains higher explanation robustness under stealthy attacks while retaining accuracy.

## 1. Introduction

Deep neural networks (DNNs) have proven their strengths in many real-world applications, including financial (Wang et al., 2020b), image retrieval (Zhou et al., 2020), and biomedical research (Hudson & Cohen, 2000). The explainability of DNNs is a fundamental requirement for establishing humans' trust and is key to further deployments in high-stake applications (Goodman & Flaxman, 2017; Pu & Chen, 2006). As human cognitive capability is limited (Saaty & Ozdemir, 2003), an explanation typically attributes a prediction to a few salient features of the input data (see Fig. 2). Among all explanation methods, saliency maps based on model gradients with respect to input data are widely adopted due to their inexpensive computation and intuitive

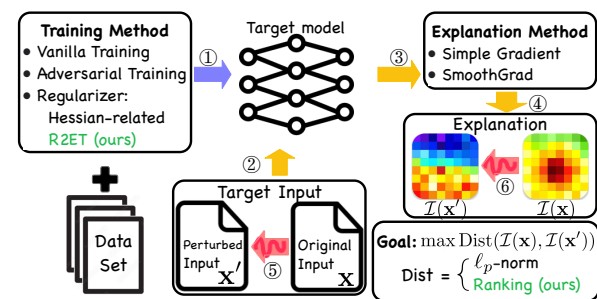

Figure 1: **Blue** (①): Model training. **Yellow** (②-④): Explanation generation for a target input. **Red** (⑤-⑥): Adversarial attacks against the explanation by manipulating the input.

interpretation (Nielsen et al., 2022).

Existing works (Dombrowski et al., 2019; Ghorbani et al., 2019; Heo et al., 2019) show that the gradients can be manipulated with unnoticeable changes in the input. They measure the explanation robustness using a certain $\ell_p$ norm, leading to the idea of minimizing the $\ell_p$ norm of the Hessian of the model against the input for robust explanations (Dombrowski et al., 2021; Wang et al., 2020c). However, as demonstrated in Fig. 2, a perturbed explanation with a small $\ell_p$ distance to the original one can have rather different top salient features, since the $\ell_p$ norm considers the importance of *all* features equally. Such inconsistency between the $\ell_p$ metric and the modus operandi of human perception can lead to mistrust of the model and the associated explanations. Alternatively, we will measure the stability of the rankings of the top salient features. Prior work on ranking robustness can be found in information retrieval (Zhou et al., 2020; 2021), though they are not applicable to stability in explainable ML, in terms of vulnerabilities, attacking objectives, theoretical analysis, and computations (see Sec. 5).

**Contributions.** We center our contributions around a novel metric called "ranking explanation thickness" that precisely measures the robustness of the top salient features. (1) Theoretically, we derive surrogate bounds of ranking explanation thickness for more efficient optimization and to reveal a fundamental limit of using Hessian norm for explanation robustness. We also disclose the equivalence between ranking explanation thickness and Adversarial Training in Eq. (7). (2) Algorithmically, based on the theoretical analysis, we propose an efficient training method, R2ET (see Fig.

[1]Lehigh University, PA, USA [2]Beijing University of Posts and Telecommunications, Beijing, China [3]University of Illinois Chicago, IL, USA [4]Carnegie Mellon University, PA, USA. Correspondence to: Sihong Xie <xiesihong1@gmail.com>.

*Workshop on Interpretable ML in Healthcare at International Conference on Machine Learning (ICML)*, Honolulu, Hawaii, USA. 2023. Copyright 2023 by the author(s).

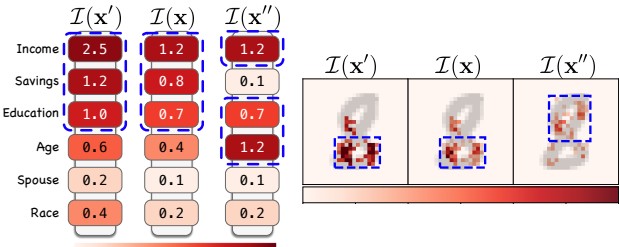

Figure 2: A smaller $\ell_p$ distance between saliency maps does not imply similar top salient features. $\mathbf{x}$ is the original input, and $\mathbf{x}'$ and $\mathbf{x}''$ are two perturbed inputs. Saliency map (explanation), denoted by $\mathcal{I}(\cdot)$, is a function of the input. The saturation of the red color indicates feature saliency, and the blue dashed boxes highlight the top salient features/regions. **Left**: $\|\mathcal{I}(\mathbf{x}) - \mathcal{I}(\mathbf{x}')\|_2 = 2.4 > 1.1 = \|\mathcal{I}(\mathbf{x}) - \mathcal{I}(\mathbf{x}'')\|_2$. However, $\mathcal{I}(\mathbf{x}')$ and $\mathcal{I}(\mathbf{x})$ have the same top-3 salient features. **Right**: $\|\mathcal{I}(\mathbf{x}) - \mathcal{I}(\mathbf{x}')\|_2 = 0.10 > 0.07 = \|\mathcal{I}(\mathbf{x}) - \mathcal{I}(\mathbf{x}'')\|_2$, but the top-50 salient features from $\mathcal{I}(\mathbf{x})$ and $\mathcal{I}(\mathbf{x}')$ have a 92% of overlap, and only 36% between $\mathcal{I}(\mathbf{x})$ and $\mathcal{I}(\mathbf{x}'')$.

1), to attain the desired robustness without costly adversarial training. R2ET optimizes the surrogate bounds to overcome the limitation of existing Hessian-based curvature smoothing approaches. (3) Experimentally, we conduct both $\ell_2$-norm attack and a ranking-based attack (see Fig. 1) to demonstrate: *i*) prior $\ell_p$-norm attack cannot manipulate as efficiently as ranking-based attacks, and Hessian-related defense algorithms do not result in robust rankings; *ii*) the high correlation between explanation thickness and robustness; *iii*) the generality and wide applications of R2ET to different neural network architectures and various data modalities, including two brain network datasets.

## 2. Preliminaries

**Saliency map explanations.** Let a classification model with parameters $\mathbf{w}$ be $f(\mathbf{x}, \mathbf{w}) : \mathcal{X} \rightarrow [0, 1]^C$, and $f(\mathbf{x}, \mathbf{w})_c$ is the probability of class $c$ for the input $\mathbf{x} \in \mathcal{X}$. We omit $\mathbf{w}$ for brevity in the sequel as $\mathbf{w}$ is fixed. Gradient-based methods (Adebayo et al., 2018; Baldassarre & Azizpour, 2019) explain $f(\mathbf{x})_c$ by the top features with the largest magnitudes. Formally, a saliency map explanation is defined as $\mathcal{I}(\mathbf{x}, c; f) = \nabla_{\mathbf{x}} f(\mathbf{x})_c$. Since the model $f$ is fixed for explanations and we fix $c$ to the *predicted* class, we use $\mathcal{I}(\mathbf{x})$ to denote $\mathcal{I}(\mathbf{x}, c; f)$, and use $\mathcal{I}(\mathbf{x})_i$ to denote the $i$-th feature's importance score.

**Threat model.** The adversary solves the following problem to find the optimal perturbation $\delta^*$ to distort the explanations without changing the predictions (Dombrowski et al., 2021):

$$\max_{\delta : \|\delta\|_2 \leq \epsilon} \text{Dist}(\mathcal{I}(\mathbf{x}), \mathcal{I}(\mathbf{x} + \delta)) \quad (1)$$
$$\text{s.t.} \quad \arg\max_c f(\mathbf{x})_c = \arg\max_c f(\mathbf{x} + \delta)_c,$$

$\delta$ is the perturbation whose $\ell_2$-norm is not larger than a given budget $\epsilon$. $\text{Dist}(\cdot, \cdot)$ evaluates how different the two

explanations are. E.g., $\text{Dist}(\cdot, \cdot)$ is inversely related to the correlation between two rankings. The constraint can make manipulating feature ranking more difficult.

## 3. Explanation Robustness via Thickness

### 3.1. Ranking explanation thickness and surrogates

**Quantify the gap.** Given a model $f$ and the associated original explanation $\mathcal{I}(\mathbf{x})$ with respect to an instance $\mathbf{x} \in \mathbb{R}^n$, we denote the *gap* between the importance of $i$-th and $j$-th features by $h(\mathbf{x}, i, j) = \mathcal{I}(\mathbf{x})_i - \mathcal{I}(\mathbf{x})_j$. Clearly, $h(\mathbf{x}, i, j) > 0$ *if and only if* when the $i$-th feature has a more significantly positive contribution to the prediction than the $j$-th feature. Although the feature importance order varies across different $\mathbf{x}$, for notation simplicity, we label the features in descending order such that the $i$-th feature is always more important than the $j$-th one, namely, $h(\mathbf{x}, i, j) > 0, \forall i < j$, with respect to the *original* input $\mathbf{x}$. This assumption will not affect the following analysis.

**Ranking robustness.** Ranking robustness (Goren et al., 2018; Zhou & Croft, 2006) measures how much the ranking changes concerning slight input perturbations. Now, we consider the perturbed input $\mathbf{x}' = \mathbf{x} + \delta$ with $\|\delta\|_2 \leq \epsilon$. Apparently, the adversary in Eq. (1) tries to flip the ranking between $(i, j)$ features such that $h(\mathbf{x}', i, j) < 0$ for some $i < j$. Meanwhile, a model with robust ranking explanation is supposed to retain the rankings between any two features, $h(\mathbf{x}', i, j) > 0, \forall i < j$. We define the *explanation* thickness by the probability of the relative ranking of the features $(i, j)$ being unchanged in a neighborhood of $\mathbf{x}$. The relevant work (Yang et al., 2020a) proposes the boundary thickness to evaluate a model's *prediction* robustness by measuring the expected distance between two level sets.

**Definition 3.1** (Local Pairwise Ranking Thickness)**.** Given a model $f$, an input $\mathbf{x} \in \mathcal{X}$ and a distribution $\mathcal{D}$ over $\mathbf{x}'$, the local pairwise ranking thickness (as a probability) of the pair of features $(i, j)$ is

$$\tilde{\Theta}(f, \mathbf{x}, \mathcal{D}, i, j) \stackrel{\text{def}}{=} \mathbb{E}_{\mathbf{x}' \sim \mathcal{D}} \left[ \int_0^1 \mathbb{1}[h(\mathbf{x}(t), i, j) \geq 0] dt \right], \quad (2)$$

where $\mathbf{x}' \sim \mathcal{D}$ is the perturbed input drawn from a neighborhood of $\mathbf{x}$. $\mathbf{x}(t) = (1 - t)\mathbf{x} + t\mathbf{x}', t \in [0, 1]$, is on the line segment connecting the sample pair $(\mathbf{x}, \mathbf{x}')$.

Clearly, $\tilde{\Theta} \leq 1$ and the equality holds when the relative importance of the $i$-th and $j$-th features is never flipped. The integration calculates the probability that the $i$-th feature is more important than $j$-th feature between $\mathbf{x}$ and $\mathbf{x}'$. The expectation considers all such probabilities where $\mathbf{x}'$ is drawn from a Uniform distribution $U(\mathbf{x}, \epsilon)$ (Wang et al., 2020c) or a Gaussian distribution $\mathcal{N}(\mathbf{x}, \sigma^2 I)$ (Smilkov et al., 2017). The expectation makes the thickness estimation more comprehensive around the neighborhood of $\mathbf{x}$. Alternatively, we

can set $\mathbf{x}'$ to an adversarial sample local to $\mathbf{x}$ (Yang et al., 2020a) to find the worst case of the thickness, and we will further discuss it in Sec. 3.2.

**Relaxation.** Due to the non-differentiability of the indicator function in Eq. (2), it is difficult to analyze and optimize the thickness efficiently. Alternatively, we remove the indicator function and define the local pairwise ranking thickness as

$$\Theta(f, \mathbf{x}, \mathcal{D}, i, j) \overset{\text{def}}{=} \mathbb{E}_{\mathbf{x}' \sim \mathcal{D}} \left[ \int_0^1 h(\mathbf{x}(t), i, j) dt \right], \quad (3)$$

which is still monotonically increasing in $h$. $\Theta(f, \mathbf{x}, \mathcal{D}, i, j)$ measures the *expected gap* between the importance score of $i$ and $j$ features. Our following analysis will focus on $\Theta$.

**Top-$k$ thickness.** Existing works in general robust ranking propose maintaining the ranking between *every* two features (Zhou et al., 2021), demanding a complexity of $\mathcal{O}(n^2)$ for $n$ features. However, as shown in Fig. 2, only the top-$k$ important features in $\mathcal{I}(\mathbf{x})$ and the robustness of their positions are more relevant to human perception of explanation. Thus, only the relative ranking between a feature from the top-$k$ and another one from the remaining ones are relevant. We define the following robustness metric that requires nearly-linear complexity $\mathcal{O}(n)$ when $k \ll n$.

**Definition 3.2** (Local Top-$k$ Ranking Thickness). Given a model $f$, an input $\mathbf{x} \in \mathcal{X}$, and a distribution $\mathcal{D}$ over $\mathbf{x}'$, the local thickness of the ranking of the top-$k$ features is

$$\Theta(f, \mathbf{x}, \mathcal{D}; k, n) \overset{\text{def}}{=} \frac{1}{m} \sum_{i=1}^{k} \sum_{j=k+1}^{n} \Theta(f, \mathbf{x}, \mathcal{D}, i, j), \quad (4)$$

where $m = k(n-k)$, and $\tilde{\Theta}(f, \mathbf{x}, \mathcal{D}; k, n)$ with an indicator function can be defined in a similar way.

### 3.2. Training for robust ranking explanations

To make attacks more difficult and thus the explanation more robust, we add $\Theta(f, \mathbf{x}, \mathcal{D}; k, n)$ as a regularizer when training $f$ on the training set $(\mathcal{X}_T, \mathcal{Y}_T)$:

$$\min_{\mathbf{w}} \mathcal{L}_{total}(f) = \mathcal{L}_{cls}(f) - \lambda \mathbb{E}_{\mathbf{x} \in \mathcal{X}_T} \left[ \Theta(f, \mathbf{x}, \mathcal{D}; k, n) \right],$$

where $\mathcal{L}_{cls}(f)$ is the empirical classification loss of $f(\mathbf{x}, \mathbf{w})$ and $\lambda > 0$ is a hyperparameter.

**A surrogate bound of explanation thickness.** Directly optimizing $\Theta$ in Eq. (4) requires $M_1 \times M_2 \times 2$ backward propagations *per* training sample *per* iteration. $M_1$ is the number of perturbed samples sampled from $U(\mathbf{x}, \epsilon)$ or $\mathcal{N}(\mathbf{x}, \sigma^2 I)$, or the number of iterations for finding the adversarial sample $\mathbf{x}'$; $M_2$ is the number of interpolations $\mathbf{x}(t)$ sampled from the line segment between $\mathbf{x}$ and $\mathbf{x}'$; and evaluating the gradient of $h(\mathbf{x}, i, j) = \mathcal{I}(\mathbf{x})_i - \mathcal{I}(\mathbf{x})_j$ requires at least 2 backward propagations (Pearlmutter, 1994). To avoid sampling $\mathbf{x}'$ and $\mathbf{x}(t)$, we derive a lower bound of $\Theta$, which requires only 2 backward propagations to maximize.

**Definition 3.3** (Locally Lipschitz continuity). A function $f$ is $L$-locally Lipschitz continuous if $\|f(\mathbf{x}) - f(\mathbf{x}')\|_2 \leq L\|\mathbf{x} - \mathbf{x}'\|_2$ holds for all $\mathbf{x}' \in \mathcal{B}_2(\mathbf{x}, \epsilon) = \{\mathbf{x}' \in \mathbb{R}^n : \|\mathbf{x} - \mathbf{x}'\|_2 \leq \epsilon\}$.

**Proposition 3.4.** (Bounds of thickness) *Given a $L$-locally Lipschitz model $f$, for some $L$, local pairwise ranking thickness $\Theta(f, \mathbf{x}, \mathcal{D}, i, j)$ is bounded by*

$$
\begin{aligned}
h(\mathbf{x}, i, j) - \epsilon * \frac{1}{2} \|H(\mathbf{x})_i - H(\mathbf{x})_j\|_2 \leq \\
\Theta(f, \mathbf{x}, \mathcal{D}, i, j) \leq h(\mathbf{x}, i, j) + \epsilon(L_i + L_j),
\end{aligned}
\quad (5)
$$

*where $H(\mathbf{x})_i$ is the derivative of $\mathcal{I}(\mathbf{x})_i$ with respect to the input $\mathbf{x}$, and $L_i = \max_{\mathbf{x}' \in \mathcal{B}_2(\mathbf{x}, \epsilon)} \|H(\mathbf{x}')_i\|_2$.*

Note that the bounds are related to $h(\mathbf{x}, i, j) = \mathcal{I}(\mathbf{x})_i - \mathcal{I}(\mathbf{x})_j$ and the Hessian of $f$. The bounds of the local top-$k$ ranking thickness could be derived similarly.

The bounds have the following implications.

- The bounds are related to $\mathbf{x}$, not $\mathbf{x}'$, and optimizing the bounds requires $M_1 \times M_2$ *fewer* times of backward propagations and frees from heavy computations in Eq. (3).

- The bounds are related to the perturbation budget $\epsilon$, but not to the distribution $\mathcal{D}$. Thus, Eq. (5) is valid for adversarial sample $\mathbf{x}'$ (Yang et al., 2020a) with $\|\delta\|_2 \leq \epsilon$ or any random distribution such as $U(\mathbf{x}, \epsilon)$ (Wang et al., 2020c).

- We reveal another motivation for minimizing Hessian norm: rather than smoothing the curvature, we aim to tighten the bounds of thickness and to ultimately maximize the thickness without dealing with neighbor sampling and line integration in Eq. (3). In particular, as $\|H(\mathbf{x})\|_2 \to 0$, we have $L_i + L_j \to 0$, $\|H_i(\mathbf{x}) - H_j(\mathbf{x})\|_2 \to 0$, and $\lim_{\|H(\mathbf{x})\|_2 \to 0} \mathbb{E}_{\mathbf{x}'} \left[ \int_0^1 h(\mathbf{x}(t), i, j) dt \right] = h(\mathbf{x}, i, j)$.

- The bounds are related to $h(\mathbf{x}, i, j) = \mathcal{I}_i(\mathbf{x}) - \mathcal{I}_j(\mathbf{x})$ and the Hessian of $f$. Thus, *only* minimizing a Hessian norm (Dombrowski et al., 2019; 2021) is insufficient for ranking explanation robustness (see Table 1).

- The most relevant prior work is (Hein & Andriushchenko, 2017). They connect the robustness of prediction $f_c$ to the ratio $\max_j \frac{f(\mathbf{x})_c - f(\mathbf{x})_j}{\|\mathcal{I}(\mathbf{x}', c) - \mathcal{I}(\mathbf{x}', j)\|_2}$, where the ratio stems from the *optimal* perturbation direction. Although we can adopt their proof to obtain a similar ratio $\frac{\mathcal{I}(\mathbf{x})_i - \mathcal{I}(\mathbf{x})_j}{\|\nabla \mathcal{I}(\mathbf{x}')_i - \nabla \mathcal{I}(\mathbf{x}')_j\|_2}$, our experiments show that the second-order term in the denominator makes the optimization less stable and the training can hardly converge (see Table 4).

Based on Prop. 3.4, simultaneously maximizing the gap and minimizing Hessian norm improve the thickness. Thus,

we have the following optimization problem for training an accurate classifier with robust feature ranking explanations:

$$\min_{\mathbf{w}} \mathcal{L}_{total}(f) = \mathcal{L}_{cls}(f) - \lambda_1 \mathbb{E}_{\mathbf{x}} \left[ \sum_{i=1}^{k} \sum_{j=k+1}^{n} h(\mathbf{x}, i, j) \right] + \lambda_2 \mathbb{E}_{\mathbf{x}} \left[ \|H(\mathbf{x})\|_2 \right],$$

(6)

where $\lambda_1, \lambda_2 \geq 0$. In this way, we optimize the gap and Hessian norm concentrating in $\mathbf{x} \in \mathcal{X}_T$ and are free from expensive sampling. When $\lambda_1 = 0$, we recover Hessian norm minimization to smooth the curvature (Dombrowski et al., 2021). When $\lambda_2 = 0$, we only increase the gaps. We call the strategy that trains $f$ using Eq. (6) *Robust Ranking Explanation via Thickness* (**R2ET**).

**Connection to adversarial training (AT).** We have following proposition based on the prior work (Xu et al., 2009):

**Proposition 3.5.** *The optimization problem in Eq. (6) is equivalent to the following min-max problem:*

$$\min_{\mathbf{w}} \max_{(\delta_{1,k+1}, \ldots, \delta_{k,n}) \in \mathcal{N}} \mathcal{L}_{cls} - \mathbb{E}_{\mathbf{x}} \left[ \sum_{i=1}^{k} \sum_{j=k+1}^{n} h(\mathbf{x} + \delta_{i,j}, i, j) \right],$$

(7)

*where $\delta_{i,j}$ is a perturbation to $\mathbf{x}$ targeting the $(i, j)$ pair of features. $\mathcal{N}$ is the feasible set of perturbations where each $\delta_{i,j}$ is independent of each other, with $\| \sum_{i,j} \delta_{i,j} \| \leq \epsilon$*

Prop. 3.5 indicates that R2ET has the same effect as the AT but uses a regularization to bypass heavy computations. Specifically, the bottleneck of the above AT is the high time complexity for finding the optimal $\delta$ for any $\mathbf{x}$, which makes the AT $M_1$ times more expensive than R2ET, where $M_1$ is the number of attack iterations per $\mathbf{x}$.

**Selecting pairs to compare.** Notice that the features in the long tail are less likely to be confused with the top-$k$ features, we set the $k$-th salient feature as the "anchor", and approximate the top-$k$ ranking thickness by $k'$ pairs of features $\sum_{i=1}^{k'} h(\mathbf{x}, k - i, k + i)$ with $\mathcal{O}(k')$ complexity. When $k' = k$, we preserve the relative rankings of the top-$2k$ features, which is named **R2ET-mm**.

## 4. Experiments

### 4.1. Experimental Settings

**Model architectures and datasets.** We adopt two types of network architectures: single-input DNNs and dual-input Siamese Networks (SNs). For single-input DNN, we use three tabular datasets: Bank (Moro et al., 2014), Adult, and COMPAS (Mothilal et al., 2020), and an image dataset CIFAR-10 (Krizhevsky et al., 2009) with ResNet. For SNs that compare two inputs, we use the image dataset MNIST (LeCun et al., 1998) and two graph datasets of brain networks: BP (Ma et al., 2019) and ADHD (Ma et al., 2016).

In BP and ADHD, each brain network comprises 82 and 116 nodes, respectively. Since the datasets are limited, we employ five-fold cross-validation, where three folds are used for training, one for validation, and one for testing. We create training sets by pairing any two training graphs. We pair any two training graphs as the training set. To simulate real medical diagnosis (by comparing a new sample with those in the database), each pair consists of a training graph and a validation (or test) graph as validation (or testing) sets.

**Evaluation metrics.** We use Precision@$k$ (P@$k$) (Ghorbani et al., 2019; Wang et al., 2020c) to quantify the similarity between two explanations before and after attacks. To ensure all the trained models have similarly good *prediction* performance, we guarantee that almost all the models have relatively high clean AUC. We further keep the adversarial AUC high and sensitivity (Xu et al., 2020) close to zero when conducting attacks. In Sec. 4.6, we report DFFOT (Serrano & Smith, 2019), comprehensiveness and sufficiency (DeYoung et al., 2019) to show that explanations from R2ET models are *faithful*.

- **Decision Flip - Fraction of Tokens (DFFOT)** (Serrano & Smith, 2019) measures the minimum fraction of important features to be removed to flip the prediction. Formally, $\min_k \frac{k}{n}$, s.t. $\arg\max_c f(\mathbf{x})_c \neq \arg\max_c f(\mathbf{x}_{[\backslash k]})_c$, where $\mathbf{x}_{[\backslash k]}$ is the perturbed input whose top-$k$ important features are removed.

- **Comprehensiveness (COMP)** (DeYoung et al., 2019) measures the changes in predictions before and after removing the most important features. Formally, $\frac{1}{\|K\|} \sum_{k \in K} |f(\mathbf{x})_c - f(\mathbf{x}_{[\backslash k]})_c|$, where $K$ is $\{1, \ldots, n\}$ for tabular data, and $\{1\% * n, 5\% * n, 10\% * n, 20\% * n, 50\% * n\}$ for images and graphs.

- **Sufficiency (SUFF)** (DeYoung et al., 2019) measures the change of predictions if only the important tokens are preserved. Formally, $\frac{1}{\|K\|} \sum_{k \in K} |f(\mathbf{x})_c - f(\mathbf{x}_{[k]})_c|$, where $\mathbf{x}_{[k]}$ is the perturbed input with only top-$k$ important features, and $K$ is set the same as the one for COMP.

**Explanation methods.** We adopt SimpleGrad as the explanation method, and similar conclusions to SimpleGrad can be drawn when adopting SmoothGrad (Smilkov et al., 2017) and Integrated Gradients (Sundararajan et al., 2017).

**Hyperparamters.** We pick $k = 8$ for three tabular data, $k = 100$ for CIFAR-10, and $k = 50$ for MNIST and graphs. We set the maximal training epoch as 300 for three tabular data, 100 for MNIST, and 10 for two graph datasets, BP and ADHD. Almost all models are guaranteed to converge within given maximal epochs, except when the regularization term weights are too large. The learning rate is set to 1e-2 for three tabular data, 1e-3 for MNIST, 1e-2 for CIFAR-10, and 1e-4 for BP and ADHD. We set $k' = k$ for R2ET

and its variants for three tabular datasets and image datasets, and $k' = 20$ for BP and ADHD. As for attacks, we conduct attacks in a PGD-style (Madry et al., 2017) for at most 1000 iterations for tabular datasets, and perturb input with a 1e-3 budget in each iteration. Thus, each input in tabular datasets can be manipulated with at most $\epsilon = 10^{-3} * 1000 = 1$ budget. The budget is set as $100 * 5e - 2$ for CIFAR-10, and $100 * 1e - 2$ for the rest. Inputs are normalized in image datasets such as CIFAR-10 and MNIST.

## 4.2. Compared Methods

We conduct two attacks in the PGD manner (Madry et al., 2017): Explanation Ranking attack (**ERAttack**) and **MSE attack**. ERAttack minimizes $\sum_{i=1}^{k} \sum_{j=k+1}^{n} h(\mathbf{x}', i, j)$ to manipulate the ranking of features in explanation $\mathcal{I}(\mathbf{x})$, and MSE attack maximizes the MSE (i.e., $\ell_2$ distance) between $\mathcal{I}(\mathbf{x})$ and $\mathcal{I}(\mathbf{x}')$. We compare the proposed defense strategy **R2ET** with the following baselines.

- **Vanilla**: provides the basic ReLU model trained without weight decay or any regularizer term.

- **Weight decay (WD) (Dombrowski et al., 2021)**: uses weight decay during training to bound Hessian norm.

- **Softplus (SP)** (Dombrowski et al., 2019; 2021): replaces ReLU with $\text{Softplus}(x; \rho) = \frac{1}{\rho} \ln(1 + e^{\rho x})$.

- **Estimated-Hessian (Est-H)** (Dombrowski et al., 2021): Hessian norm as the regularizer, which is estimated by the finite difference (Moosavi-Dezfooli et al., 2019): $\|\frac{\nabla f(\mathbf{x}+\kappa\mathbf{v})-\nabla f(\mathbf{x})}{\kappa}\|_2$, where $\kappa \ll 1$, $\mathbf{v} = \frac{\text{sign}(\nabla f(\mathbf{x}))}{||\text{sign}(\nabla f(\mathbf{x}))||_2}$. It can be considered an ablation variant of R2ET ($\lambda_1 = 0$).

- **Exact-Hessian (Exact-H)**: the exact Hessian norm is used as the regularizer.

- **SSR** (Wang et al., 2020c): sets the largest eigenvalue of the Hessian matrix as the regularizer.

- **Adversarial Training (AT) for robust prediction** (Huang et al., 2016; Wong et al., 2020): find $f$ by $\min_f \sum_{(\mathbf{x},y)\in(\mathcal{X}_T, \mathcal{Y}_T)} (\mathcal{L}_{cls}(f; \mathbf{x} + \delta^*, y))$, where $\delta^* = \arg\max_\delta - \sum_{i=1}^{k} \sum_{j=k+1}^{n} h(\mathbf{x} + \delta, i, j)$.

- **R2ET-mm**: selects *multiple* distinct $i, j$ with *minimal* $h(\mathbf{x}, i, j)$ as discussed in Sec. 3.2.

- **R2ET$_{\setminus H}$** and **R2ET-mm$_{\setminus H}$**: They are the ablation variants of **R2ET** and **R2ET-mm**, respectively, without optimizing the Hessian-related term in Eq. (6) ($\lambda_2 = 0$).

## 4.3. Overall robustness results

**Attackability of ranking-based explanation.** Table 1 reports P@$k$ under ERAttack and MSE attacks for every model on all datasets. We observe that more than 50% of

models achieve at least 90% P@$k$ under MSE attacks, concluding that MSE attack cannot effectively alter the rankings of salient features, even without extra defense (row Vanilla). The ineffective attack method can give a false impression of explanation robustness, and a stronger attack is needed. ERAttack, on the other hand, can remove more salient features from the top-$k$ positions for most models and datasets, leading to significantly lower P@$k$ values than MSE attack. For example, about 50% of the top 50 features are out of top positions under ERAttack on average on ADHD, while less than 20% of the top 50 features drop out of the top under the MSE attack.

**Effectiveness of ranking thickness against ERAttacks.** We compare the performance of different defense strategies against ERAttack, which is more effective than MSE attacks. Similar conclusion can be made with MSE attacks case. First, R2ET and its variants achieve the best (highest) top-$k$ explanation robustness for most datasets, indicating R2ET methods' superiority for preserving the top salient features. Second, it is counter-intuitive that R2ET$_{\setminus H}$, as an ablation version of R2ET, outperforms R2ET on Adult and Bank. The reason is that R2ET$_{\setminus H}$ has a better ranking thickness on these datasets than R2ET (see Fig. 3 in Sec. 4.4). We conjecture that the number of features in the dataset can serve as a straightforward and intuitive indicator to determine the potential performance of R2ET or R2ET$_{\setminus H}$ in practical scenarios. Specifically, in cases where the number of features is *small*, such as in the Adult, Bank, and COMPAS datasets, it becomes easier to restrict the relative rankings among a limited set of features. As a result, both R2ET$_{\setminus H}$ and R2ET-mm$_{\setminus H}$ demonstrate good performance. In this case, reducing the Hessian norm diminishes the gap between features and adversely affects robustness, as observed when comparing Vanilla and Est-H or Exact-H. Conversely, when the number of features is *large*, it becomes significantly more challenging to maintain all the rankings solely by expanding the gaps between features, which is the approach taken by R2ET$_{\setminus H}$ and R2ET-mm$_{\setminus H}$. Due to the sheer number of features involved, R2ET, which simultaneously expands the gaps and minimizes the Hessian norm, has the potential to outperform the other methods. The theoretical discussion is disclosed in Sec. 3.2. Lastly, we consider the baselines that strive for a smoother curvature without considering the absolute gaps among feature importance, including WD, SP, Est-H, Exact-H, and SSR. Overall, their performance is unstable across datasets (SP on COMPAS and Est-H on BP). However, the best performers always have the largest thickness as demonstrated in Sec. 4.4. The above observations show that ranking thickness is a more fundamental measurement of ranking robustness. Besides that, Est-H, Exact-H, and SSR smooth the curvature by adding Hessian-related terms, either Hessian norm or the maximal eigenvalue of Hessian, and they perform similarly.

Based on their performance in Table 1, optimizing Hessian norm or its relevant terms solely is not sufficient to improve the ranking thickness. Since it is extremely expensive to compute the exact Hessian norm and its eigenvalues, both Exact-H and SSR are inapplicable to MNIST, CIFAR-10, ADHD and BP. We adopt "fast"-AT (Wong et al., 2020) for AT baseline, where the inner maximization is solved by a single-step attack to balance the training time and robust performance. "Fast"-AT, however, suffers from unstable robust performance as studied in (Li et al., 2020), and cannot perform well in most datasets.

### 4.4. Analysis on Ranking Thickness and Existing Metric

As shown in Table 1, R2ET and its variants are the winners on almost all the datasets except COMPAS and BP, and R2ET$_{\setminus H}$ and R2ET-mm$_{\setminus H}$ outperform their counterparts (R2ET and R2ET-mm) on Adult and Bank. We aim to identify the essential factor that helps a method outperform others. We will use the number of iterations to reach a successful attack where *any* salient feature is swapped out of the top $k$ positions of the original explanation.

To explore the correlations among metrics, for each sample $\mathbf{x}$, we collect the number of iterations to the first flip under ERAttack, Hessian norm, and thickness. Each dot in Fig. 3 indicates one sample $\mathbf{x}$, and the correlation coefficient between metrics is shown in each subplot. Compared with Hessian norm, thickness is significantly more correlated with the number of iterations to the first flip. It indicates that Hessian norm is *not* a strong indicator of explanation ranking robustness. Recall the discussion in Sec 3.2, minimizing Hessian norm helps tighten the bounds. However, optimizing Hessian norm *solely* may only marginally contribute to the ranking robustness, and it is consistent with the observations in Table 1, where Est-H and Exact-H do not perform well most of the time. As thickness is a precise measurement for the robustness of explanations, we further report the *model-level* thickness, an average of thickness over all samples, in Table 2. It could answer why R2ET cannot always outperform baselines: On Adult and COMPAS, R2ET$_{\setminus H}$ has a higher thickness (0.999 and 0.973, resp.) than R2ET (0.997 and 0.987, resp.). We have similar observations for BP, where Est-H has a higher thickness (0.93563) than R2ET-mm (0.93561).

### 4.5. Case study: saliency maps visualization

Besides the robustness of ranking explanations, we hope optimizing thickness can lead to models that identify the ground-truth important features (e.g., pixels covering the digits), since a model that robustly uses irrelevant features is not useful in practice. MNIST and CIFAR-10 provide suitable testbeds to visually evaluate whether R2ET forces the model to use irrelevant features.

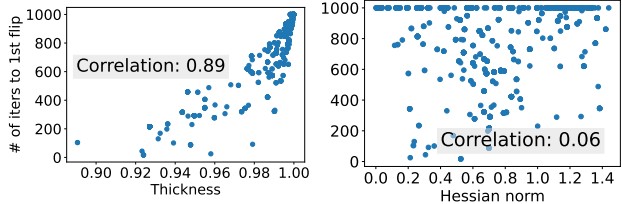

Figure 3: Correlation between the number of iterations to first flip, and ranking thickness (left) and Hessian norm (right) for R2ET model on COMPAS.

In Fig.4, we conduct ERAttack against models trained with different strategies. In MNIST, Vanilla performs poorly, with about $30\% \sim 50\%$ of the top 50 important features fell out of top positions under ERAttack. WD improves P@$k$ to 56% and 68% compared with Vanilla in the example. For R2ET and R2ET-mm, it is difficult to find visible change in the top salient features before and after ERAttack, and P@$k$ of both methods are greater than 90%. We can also see that the top 50 salient pixels used by Vanilla and WD do not highlight the spatial patterns of the digits. However, the explanations show that the top 50 important features used by R2ET and R2ET-mm encode recognizable spatial patterns of the digits. We have similar conclusions in CIFAR-10. Specifically, take the explanations on the ship in Fig. 4 bottom as an example. Vanilla and WD perform badly in terms of P@$k$, while R2ET and R2ET-mm achieve around as high as 90% P@$k$. All four models make correct predictions due to a similar region (front hull of the ship). However, ERAttack manipulates the explanations of Vanilla and WD to include another region (wheelhouse of the ship) while the key regions of the explanations of R2ET and R2ET-mm under attacks remain the same. The wheelhouse may be one reason for classifying the image to ship, but the inconsistency of explanations due to imperceptible perturbations raises confusion and mistrust.

### 4.6. Explanation Faithfulness

To ensure the *faithfulness* of explanations for R2ET models, we evaluate their faithfulness using three widely accepted metrics. The results, presented in Table 3, demonstrate that the explanations provided by R2ET models are on par with, if not superior to, other models in terms of faithfulness.

### 4.7. Sensitivity Analysis

**Impacts of pretrain / retrain.** We explore how good are these methods when applying them in the **retrain** schema. In previous experiments, all models are trained from random states. We now retrain the Vanilla models with these methods for 10 epochs at most. Since the Vanilla model has already converged and reached a good cAUC, we assume that the Vanilla model's explanation ranking is an excellent reference, and thus these robust methods try to maintain the

Table 1: P@$k$ (shown in percentage) of different robust models (rows) under **ERAttack / MSE attack**. $k = 8$ for the first three dataset, and $k = 50$ for the rest. Numbers in **bold** indicate the winner on the dataset, and numbers indicate the runner-up. ($*$ Est-H has 4.6% and 3.9% lower clean AUC than R2ET-mm under two attacks, respectively, and is less useful in practice.)

| Method | Adult | Bank | COMPAS | MNIST | CIFAR-10 | ADHD | BP |
|---|---|---|---|---|---|---|---|
| Vanilla | 87.6 / 87.7 | 83.0 / 94.0 | 84.2 / 99.7 | 59.0 / 64.0 | 66.5 / 68.3 | 45.5 / 81.1 | 69.4 / 88.9 |
| WD | 91.7 / 91.8 | 82.4 / 85.9 | 87.7 / 99.4 | 59.1 / 64.8 | 64.2 / 65.6 | 47.6 / 79.4 | 69.4 / 88.6 |
| SP | 97.4 / 97.5 | 95.4 / 95.5 | **99.5 / 100.0** | 62.9 / 66.9 | 67.2 / 71.9 | 42.5 / 81.3 | 68.7 / 90.1 |
| Est-H | 87.1 / 87.2 | 78.4 / 81.8 | 82.6 / 97.7 | 85.2 / 90.2 | 77.1 / 78.7 | 58.2 / 83.7 | **75.0**$^*$ / **91.4**$^*$ |
| Exact-H | 89.6 / 89.7 | 81.9 / 85.6 | 77.2 / 96.0 | - / - | - / - | - / - | - / - |
| SSR | 91.2 / 92.6 | 76.3 / 84.5 | 82.1 / 97.2 | - / - | - / - | - / - | - / - |
| AT | 68.4 / 91.4 | 80.0 / 88.4 | 84.2 / 90.5 | 56.0 / 63.9 | 61.6 / 66.8 | 59.4 / 81.0 | 72.0 / 89.0 |
| R2ET$_{\setminus H}$ | **97.5 / 97.7** | **100.0 / 100.0** | 91.0 / 99.2 | 82.8 / 89.7 | 67.3 / 72.2 | 60.7 / 86.8 | 70.9 / 89.5 |
| R2ET-mm$_{\setminus H}$ | 93.5 / 93.6 | 95.8 / 98.2 | 95.3 / 97.2 | 81.6 / 89.7 | 77.7 / **79.4** | 64.2 / 88.8 | 72.4 / 91.0 |
| R2ET | 92.1 / 92.7 | 80.4 / 90.5 | 92.0 / 99.9 | **85.7** / 90.8 | 75.0 / 77.4 | **71.6 / 91.3** | 71.5 / 89.9 |
| R2ET-mm | 87.8 / 87.9 | 75.1 / 85.4 | 82.1 / 98.4 | 85.3 / **91.4** | **78.0** / 79.1 | 58.8 / 87.5 | 73.8 / 91.1 |

Table 2: **P@$k$** (shown in percentage) of different robust models (rows) under ERAttack and **model-level thickness**.

| Method | Adult | Bank | COMPAS | MNIST | ADHD | BP |
|---|---|---|---|---|---|---|
| Vanilla | 87.6 / 0.9889 | 83.0 / 0.9692 | 84.2 / 0.9533 | 59.0 / 0.9725 | 45.5 / 0.9261 | 69.4 / 0.9282 |
| WD | 91.7 / 0.9960 | 82.4 / 0.9568 | 87.7 / 0.9769 | 59.1 / 0.9732 | 47.6 / 0.9343 | 69.4 / 0.9298 |
| SP | 97.4 / 0.9983 | 95.4 / 0.9978 | **99.5 / 0.9999** | 62.9 / 0.9771 | 42.5 / 0.9316 | 68.7 / 0.9300 |
| Est-H | 87.1 / 0.9875 | 78.4 / 0.9583 | 82.6 / 0.9557 | 85.2 / 0.9948 | 58.2 / 0.9578 | **75.0 / 0.9356** |
| Exact-H | 89.6 / 0.9932 | 81.9 / 0.9521 | 77.2 / 0.9382 | - / - | - / - | - / - |
| SSR | 91.2 / 0.9934 | 76.3 / 0.9370 | 82.1 / 0.9549 | - / - | - / - | - / - |
| AT | 68.4 / 0.9372 | 80.0 / 0.9473 | 84.2 / 0.9168 | 56.0 / 0.9639 | 59.4 / 0.9597 | 72.0 / 0.9342 |
| R2ET$_{\setminus H}$ | **97.5 / 0.9989** | **100.0 / 1.0000** | 91.0 / 0.9727 | 82.8 / **0.9949** | 60.7 / 0.9588 | 70.9 / 0.9271 |
| R2ET-mm$_{\setminus H}$ | 93.5 / 0.9963 | 95.8 / 0.9874 | 95.3 / 0.9906 | 81.6 / 0.9942 | 64.2 / 0.9622 | 72.4 / 0.9342 |
| R2ET | 92.1 / 0.9970 | 80.4 / 0.9344 | 92.0 / 0.9865 | **85.7 / 0.9949** | **71.6 / 0.9731** | 71.5 / 0.9296 |
| R2ET-mm | 87.8 / 0.9943 | 75.1 / 0.9102 | 82.1 / 0.9544 | 85.3 / 0.9948 | 58.8 / 0.9588 | 73.8 / **0.9356** |

Table 3: Faithfulness of explanations evaluated by DFFOT ($\downarrow$) / COMP ($\uparrow$) / SUFF ($\downarrow$).

| Method | Adult | Bank | COMPAS | MNIST | ADHD | BP |
|---|---|---|---|---|---|---|
| Vanilla | 0.24 / 0.43 / 0.18 | 0.23 / 0.14 / 0.04 | **0.17** / 0.37 / 0.14 | 0.37 / 0.16 / 0.23 | 0.51 / 0.05 / 0.28 | 0.40 / 0.06 / 0.29 |
| WD | 0.45 / 0.47 / 0.23 | 0.36 / 0.27 / 0.07 | 0.29 / 0.41 / 0.18 | 0.37 / 0.16 / 0.22 | 0.49 / 0.06 / 0.27 | **0.35** / 0.05 / 0.33 |
| SP | 0.43 / 0.47 / 0.25 | 0.35 / 0.31 / 0.07 | 0.29 / **0.45** / 0.18 | 0.38 / 0.15 / 0.22 | **0.30** / 0.10 / 0.34 | 0.38 / 0.06 / 0.30 |
| Est-H | 0.44 / 0.44 / 0.24 | 0.18 / 0.21 / 0.06 | 0.27 / 0.42 / 0.17 | 0.23 / 0.24 / **0.18** | 0.59 / 0.04 / 0.26 | 0.45 / 0.05 / **0.24** |
| Exact-H | 0.43 / 0.46 / 0.23 | 0.19 / 0.14 / 0.04 | 0.30 / 0.40 / 0.18 | - / - / - | - / - / - | - / - / - |
| SSR | 0.54 / 0.39 / 0.21 | 0.46 / 0.04 / **0.01** | 0.32 / 0.43 / 0.18 | - / - / - | - / - / - | - / - / - |
| AT | 0.16 / 0.14 / **0.08** | 0.19 / 0.10 / 0.03 | 0.24 / 0.10 / **0.07** | 0.40 / 0.12 / 0.28 | 0.35 / 0.10 / 0.26 | 0.46 / 0.06 / 0.25 |
| R2ET$_{\setminus H}$ | **0.13** / **0.50** / 0.14 | 0.34 / 0.32 / 0.10 | **0.17** / 0.40 / 0.17 | 0.23 / 0.22 / 0.19 | 0.38 / 0.13 / 0.37 | 0.43 / **0.07** / 0.29 |
| R2ET-mm$_{\setminus H}$ | 0.42 / 0.47 / 0.22 | 0.34 / **0.41** / 0.14 | 0.25 / 0.42 / 0.17 | 0.25 / 0.22 / 0.21 | 0.37 / **0.17** / 0.37 | 0.42 / **0.07** / 0.29 |
| R2ET | 0.32 / 0.46 / 0.19 | **0.11** / 0.24 / 0.07 | 0.27 / 0.39 / 0.17 | **0.18** / **0.26** / 0.23 | 0.48 / 0.12 / 0.26 | 0.42 / **0.07** / 0.29 |
| R2ET-mm | 0.38 / 0.48 / 0.20 | 0.12 / 0.21 / 0.08 | 0.28 / 0.44 / 0.15 | 0.19 / **0.26** / 0.22 | 0.50 / 0.04 / **0.25** | 0.45 / 0.05 / 0.29 |

Vanilla model's rankings. Thus, we will terminate the re-training phase if P@$k$ between Vanilla model's explanation ranking and the retrain model's ranking significantly drops, or the retrain model's cAUC drops a lot.

Table 4 presents the results for comparing two training schemas. Since the baseline SP changes the models' structure (activation function), we do not consider it here. Instead, we consider another baseline, CL (Hein & Andriushchenko, 2017), and it is adopted here due to much fewer re-training epochs. More details for CL can be found in Sec. 3.2. Be-

sides that, *none* of retrain models by Exact-H and SSR can maintain Vanilla model's explanation rankings and cAUC at the same time on Bank, and thus both are not applicable.

## 5. Related Work

**Explainable machine learning and explanation robustness.** Recent post-hoc explanation methods for deep networks can be categorized into gradient-based (Zhou et al., 2016; Selvaraju et al., 2017; Baldassarre & Azizpour, 2019;

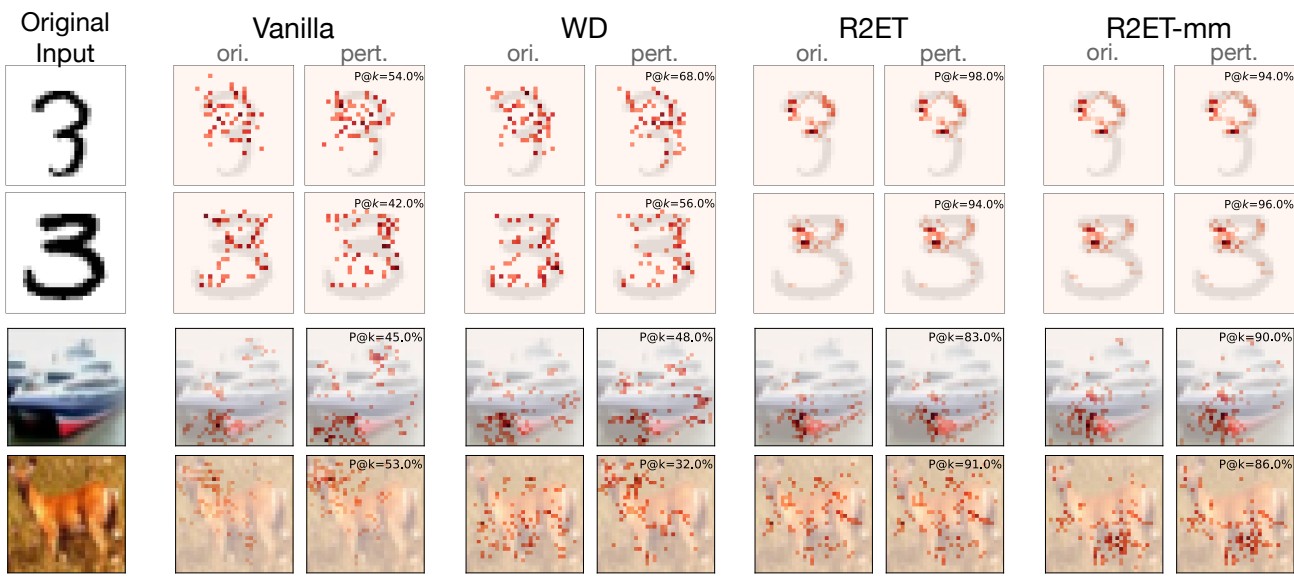

Figure 4: **Top**: With the Siamese network that compares two images from MNIST, saliency maps for the original and perturbed pairs of input images under ERAttack against different robust methods. The examples of a pair of images are from the same class (digit *3*). **Bottom**: Two images from CIFAR-10, one from class *ship* and another from class *deer*. The top $k$ (50 for MNIST and 100 for CIFAR-10) salient pixels are highlighted, and darker colors indicate higher importance. The robustness metric P@$k$ is printed within each subplot.

Table 4: P@$k$ (shown in percentage) of models under ER-Attack when the models are trained from a **random state** or **retrained** from the Vanilla models.

| Method $\parallel$ | Adult | Bank | COMPAS |
|---|---|---|---|
| Vanilla | 87.6 / 87.6 | 83.0 / 83.0 | 84.2 / 84.2 |
| WD | 91.7 / 88.3 | 82.4 / 82.1 | 87.7 / 82.7 |
| CL | - / 93.1 | - / 100.0 | - / 87.1 |
| Est-H | 87.1 / 92.1 | 78.4 / 85.2 | 82.6 / 85.1 |
| Exact-H | 89.6 / 88.7 | 81.9 / - | 77.2 / 87.0 |
| SSR | 91.2 / 88.7 | 76.3 / - | 82.1 / 86.1 |
| R2ET$_{\backslash H}$ | 97.5 / 100.0 | 100.0 / 100.0 | 91.9 / 97.8 |
| R2ET-mm$_{\backslash H}$ | 93.5 / 100.0 | 95.8 / 98.3 | 95.3 / 95.6 |
| R2ET | 92.1 / 92.6 | 80.4 / 86.2 | 92.0 / 85.1 |
| R2ET-mm | 87.8 / 91.6 | 75.1 / 86.2 | 82.1 / 87.4 |

Smilkov et al., 2017; Sundararajan et al., 2017; Shrikumar et al., 2017), surrogate model based (Ribeiro et al., 2016; Huang et al., 2022), Shapley values (Lundberg & Lee, 2017; Liu et al., 2020; Ancona et al., 2019), and causality (Pearl, 2018; Chattopadhyay et al., 2019). Although gradient-based methods are widely used (Nielsen et al., 2022), they are found to lack robustness against small perturbations (Ghorbani et al., 2019; Heo et al., 2019). Some works (Chen et al., 2019; Singh et al., 2020; Ivankay et al., 2020; Wang & Kong, 2022; Sarkar et al., 2021; Upadhyay et al., 2021) propose to improve the explanation robustness by adversarial training (AT). To bypass the high time complexity of AT, some works propose replacing ReLU function with softplus (Dombrowski et al., 2021), training with weight decay (Dombrowski et al., 2019), and incorporating gradient- and Hessian-related terms as regularizers (Wang et al., 2020c; Wicker et al., 2023). Some works propose *explanation* methods, rather than *training* methods, to enhance explanation robustness (Smilkov et al., 2017; Lu et al., 2021; Liu et al., 2022; Chen et al., 2021; Manupriya et al., 2022; Rieger & Hansen, 2020). Besides, many works (Madry et al., 2017; Tu et al., 2019; Roth et al., 2020; Yang et al., 2020a;b; Deng et al., 2021; Tsipras et al., 2018; Wen et al., 2020; Zhang et al., 2019) for adversarial robustness focus on *prediction* robustness, instead of *explanation* robustness.

**Ranking robustness and manipulations.** The ranking robustness is well-studied in information retrieval (IR), in terms of "noise" (Zhou & Croft, 2006) and adversarial attacks (Goren et al., 2018). In (Wang et al., 2020a), authors attacked image ranking by maximizing (minimizing, resp.) the similarity of mismatched (matched resp.) image pairs. Black-box attacks (Li et al., 2021) and targeted manipulations (Tolias et al., 2019) on rankings are also studied. Attacks on IR and explanations are different in three aspects. 1) Vulnerabilities: IR has queries and candidates that can be attacked (Zhou et al., 2020; 2021), while we focus on attacking the gradient of classifiers via input manipulation. 2) Attacking objectives: On IR, authors either manipulate the ranking of *one* single candidate, or manipulate query to distort the ranking of candidates. We aim to swap *any* pairs of salient and non-salient features. 3) Computations: explanations are defined by gradient or its variants and studying their robustness requires second or higher-order derivatives. The cost of higher-order derivatives motivates us to design a surrogate regularizer to bypass the costly computations.

## 6. Conclusion

We proposed "*explanation ranking thickness*" to measure the robustness of the top-ranked salient explaining features to align with human cognitive capability when interpreting a classifier's predictions. We provided theoretical insights, including surrogate bounds of the thickness, the connection between thickness and a min-max optimization problem, and a global convergence rate of a constrained multi-objective attacking algorithm against the thickness. The theory leads to a well-justified optimization problem and an efficient training algorithm *R2ET*. On 7 datasets (vectors, images, and graphs) and with 2 neural network architectures, we compared 7 state-of-the-art baselines and 3 variants of R2ET, and consistently confirmed that explanation ranking thickness is indeed a strong indicator of the stability of top salient features. In the future, we plan to consider it in the natural language processing problems and to explore R2ET with larger language models.

## Acknowledgements

Chao Chen and Sihong Xie were supported in part by the National Science Foundation under NSF Grants IIS-1909879, CNS-1931042, IIS-2008155, and IIS-2145922. Chenghua Guo and Xi Zhang were supported by the Natural Science Foundation of China (No. 61976026) and the CAAI-Huawei MindSpore Open Fund.

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
