# OpenReview forum: "Robust Ranking Explanations"
_ICML.cc/2023/Workshop/IMLH — IMLH 2023 Poster_

### Official Review · Reviewer_UbKu · 2023-06-16
**Actually, my research direction is different from that of this paper, and I am not particularly familiar with the research of this paper.**

**Rating:** 6
**Confidence:** 1

**Review:**

The paper presents a well-written definition of explanation thickness for measuring salient features ranking stability. The proposed R2ET algorithm can efficiently maximize the thickness and anchor top salient features. Additionally, the authors establish a connection between R2ET and adversarial training.

---

### Official Review · Reviewer_wJH4 · 2023-06-17
**Interesting paper with some minor issues.**

**Rating:** 6
**Confidence:** 2

**Review:**

This paper focus on robust explanations of machine learning models. Instead of using Lp norm to measure the robustness, this paper defines explanation thickness. Then, they derive tractable surrogate bounds of the thickness to design the proposed method. Conducted solid experiments, the proposed method attains higher explanation robustness under stealthy attacks while retaining accuracy.

Pros: the proposed method looks reasonable to me, and reasonably shows good results. This paper proposed both theoretical and experimental materials.

I don't see obvious cons. It might be helpful if also testing the proposed method on ImageNet. It might be also interesting to see the performance against some recent attacks such as Auto Attack. Besides, the presentation can also be improved.

---

### Official Review · Reviewer_dfWH · 2023-06-17
**Robust Ranking Explanations**

**Rating:** 8
**Confidence:** 4

**Review:**

## Summary
The paper proposes a novel method for generating robust ranking explanations for machine learning models, especially those based on gradient-based saliency maps. The authors define a metric called explanation thickness to measure the stability of the top salient features under adversarial perturbations, and derive a tractable surrogate bound of the thickness to design an efficient algorithm called R2ET. The authors also prove a connection between R2ET and adversarial training, and conduct extensive experiments on various datasets and models to demonstrate the effectiveness and superiority of R2ET over existing methods.

## Strengths
•	The paper addresses an important and timely problem of robust explainability for machine learning models, which is crucial for establishing human trust and deploying models in high-stake applications.

•	The paper introduces a novel and intuitive metric called explanation thickness to quantify the robustness of ranking explanations, which is more aligned with human perception than existing lp-norm based metrics.

•	The paper derives a tractable surrogate bound of the explanation thickness and proposes an efficient algorithm called R2ET to optimize it. The paper also proves a theoretical connection between R2ET and adversarial training, which sheds light on the underlying mechanism of R2ET.

•	The paper conducts extensive experiments on various datasets and models, including brain networks, to demonstrate the effectiveness and superiority of R2ET over existing methods. The paper also evaluates the faithfulness of the explanations generated by R2ET using several widely accepted metrics.

## Weaknesses

•	The paper lacks a clear and comprehensive literature review on related work on robust explainability and ranking stability. The paper only briefly mentions some existing methods in the introduction and related work sections, but does not provide a detailed comparison or discussion on their advantages and disadvantages, or how they differ from R2ET.

•	The paper does not provide sufficient details or justification on some of the design choices or assumptions made in the proposed method. For example, the paper does not explain why k′ pairs are sufficient to approximate the top-k ranking thickness, or how to choose k′ in practice. The paper also does not justify why using Hessian norm as a regularizer is better than other alternatives, such as gradient norm or Lipschitz constant.

---

### Official Review · Reviewer_Lr8m · 2023-06-17

**Rating:** 5
**Confidence:** 3

**Review:**

Long paper (8 pages)

Summary:
This paper proposes a new method to construct more robust input imortance explanations for neural networks based on making sure the importance ranking doesnt change.

Strengths:
- Good justification for focus on ranking stability instead of lp norm
- Mostly clearly written
- Good baselines and comparisons

Weaknesses:
- No accuracy/auc numbers for diff methods. Major flaw  needs to shown, its trivial to make robust models if accuracy suffers a lot. Should include coean and attacked acc for all methods
- Inconclusive results, unclear if this method/which version would be better on a new task
- Weird formatting of perturbation size, what is the input scale? For cifar should normalize input to 0-1 and report as 5/255 for example.

---

### Meta-Review · Program_Chairs · 2023-06-19

**Recommendation:** Accept (Poster)
**Confidence:** 4

**Metareview:**

The paper proposes a novel generating robust ranking explanations method and demonstrate on medical data. All the reviewers appreciate the focus of this work and the presentation of the manuscript. The authors are encouraged to incorporate the concerns and clarify the confusions in the final version.

---

### Decision · Program_Chairs · 2023-06-20

Accept (Poster)